# A Highly Sensitive Chitosan-Based SERS Sensor for the Trace Detection of a Model Cationic Dye

**DOI:** 10.3390/ijms25179327

**Published:** 2024-08-28

**Authors:** Bahareh Vafakish, Lee D. Wilson

**Affiliations:** Department of Chemistry, University of Saskatchewan, 110 Science Place, Thorvaldson Building, Saskatoon, SK S7N 5C9, Canada; baharheh.vafakish@usask.ca

**Keywords:** surface-enhanced Raman spectroscopy (SERS), methylene blue, silver nanoparticles, chitosan, dye adsorption, structure–function relationships

## Abstract

The rapid detection of contaminants in water resources is vital for safeguarding the environment, where the use of eco-friendly materials for water monitoring technologies has become increasingly prioritized. In this context, the role of biocomposites in the development of a SERS sensor is reported in this study. Grafted chitosan was employed as a matrix support for Ag nanoparticles (NPs) for the surface-enhanced Raman spectroscopy (SERS). Chitosan (CS) was decorated with thiol and carboxylic acid groups by incorporating S-acetyl mercaptosuccinic anhydride (SAMSA) to yield CS-SAMSA. Then, Ag NPs were immobilized onto the CS-SAMSA (Ag@CS-SAMSA) and characterized by spectral methods (IR, Raman, NIR, solid state ^13^C NMR with CP-MAS, XPS, and TEM). Ag@CS-SAMSA was evaluated as a substrate for SERS, where methylene blue (MB) was used as a model dye adsorbate. The Ag@CS-SAMSA sensor demonstrated a high sensitivity (with an enhancement factor ca. 10^8^) and reusability over three cycles, with acceptable reproducibility and storage stability. The Raman imaging revealed a large SERS effect, whereas the MB detection varied from 1–100 μM. The limits of detection (LOD) and quantitation (LOQ) of the biocomposite sensor were characterized, revealing properties that rival current *state-of-the-art* systems. The dye adsorption profiles were studied via SERS by fitting the isotherm results with the Hill model to yield the ΔG°_ads_ for the adsorption process. This research demonstrates a sustainable dual-function biocomposite with tailored adsorption and sensing properties suitable for potential utility in advanced water treatment technology and environmental monitoring applications.

## 1. Introduction

The controlled removal and detection of industrial contaminants in water present a significant environmental challenge. Detection-based methods that employ adsorbent materials have gained recognition for their dual-function properties that afford the controlled uptake and monitoring of these pollutants. Adsorption-based techniques are praised for their cost-effectiveness, versatility, ease of use, and minimal reliance on additional chemicals [1,2]. SERS is a promising and versatile analytical technique that has received tremendous attention due to its utility for the detection of many organic, inorganic, and biological species without the need for complicated sample preparation [3,4]. Since the discovery of SERS in the early 1970s, it has evolved into an accurate, sensitive, and non-destructive method [5], which reveals the unique potential for contaminant analyses, especially in aqueous media [6]. SERS takes advantage of the plasmonic resonance effect of noble metal nanoparticles (NPs) like Au and Ag. In turn, it is possible to enhance the sensitivity of the plasmonic signals by several orders of magnitude if the excitation wavelength is close to the oscillation of the conduction electrons of the NPs, referred to as the LSPR [7]. Two main mechanisms for the SERS effect include electromagnetic enhancement (EM) and the charge transfer (CT) complex formation [8]. The EM mechanism takes place when the conduction electrons of the NPs oscillate with the excitation wavelength and generate a giant electromagnetic field, which affects the spectral probe in the proximity of these particles, which depends on their distance and orientation [9]. The second mechanism is a short-range tool that enhances the Raman signal by an electronic charge transfer between the target species and the metallic NPs [10]. For considerable signal enhancement, this is an electromagnetic process that occurs when a coherent oscillation of free electrons with the excitation light occurs, which also induces an electric field around the NPs. Hence, laser light amplification takes place in small regions known as hotspots [11], where the field that the target analyte experiences in the proximity of a hotspot is typically stronger. The effect is in contrast with other target analytes that yield stronger Raman scattering [12].

The probe molecules will interact with the NPs at a short distance via chemisorption or physisorption [13]. Studies of the adsorption properties onto SERS substrates provide insight into the molecular interaction effects, which contribute to enhanced adsorbate–substrate interactions that yield a more effective SERS substrate with an enhanced selectivity and sensitivity [14].

Generally, metals such as gold, silver, or their mixtures are embedded into suitable matrices and employed as SERS substrates [15]. The matrix is the stabilizing and supporting media for immobilizing the metal NPs to prevent aggregation and to prolong storage life [16]. Chitosan is a highly abundant biopolymer derived from chitin, which represents a promising matrix for the preparation of SERS substrates [17]. Recent SERS studies that employ chitosan-based substrates highlight various benefits such as biodegradability, low cost, and the sustainable nature of such materials [18,19]. Unmodified chitosan bears abundant functional groups (e.g., hydroxyl, amine, and amide), which can serve as potential chelation sites for metal NPs, whereas modified chitosan offers a potentially superior matrix alternative [20]. The surface grafting of high affinity functional groups with chelator properties anchors metal particles to the surface of chitosan to enable SERS enhancement [21,22,23]. The surface grafting of thiol (-SH) groups onto chitosan offers an effective chelation of the metal NPs with a strong stabilizing biopolymer matrix to yield promising substrates for SERS [24].

We propose an innovative strategy by functionalizing chitosan with mercaptosuccinic acid along with the immobilization of Ag NPs to yield an effective substrate for enhanced Raman scattering, particularly for the detection of cationic dyes. To the best of our knowledge, this is the *first example* of grafting chitosan with S-acetyl mercaptosuccinic acid (SAMSA) that employs such a facile and low-cost synthetic route. The fabricated matrix immobilized with Ag NPs was characterized and applied to SERS studies for the analytical detection of MB, a well-known cationic dye that is widely used in diverse industries (e.g., cosmetics, pigments, and pharmaceuticals) [25,26,27]. Raman micro-imaging was used to evaluate whether the distribution of the Ag NPs was homogenous over the substrate surface, where the strong enhancement of the SERS signal with an enhancement factor (EF) ca. 10^8^ and a limits of detection (LOD) ca. 1.6 nM was demonstrated for methylene blue (MB). These EF and LOD values are noteworthy, as compared with other related systems from the literature [28,29]. Furthermore, the SERS performance was optimized by changing the content of Ag NPs on the substrate. The adsorption behavior of MB on the SERS substrate was fitted by the Hill isotherm model, where cooperative MB dye binding occurs on the Ag NPs’ surface. This work demonstrates that the reproducibility and regeneration of the biocomposite substrate contributes to an inexpensive and durable platform sensor material for SERS analysis due to the unique dual-function properties of this hybrid biopolymer substrate. This study makes a substantial contribution to the understanding of chitosan biocomposites and their structure–adsorption relationships, offering valuable insights for adsorption and sensor technologies. The findings hold promise for advancements in water treatment applications, particularly for the trace detection of cationic dyes in environmental monitoring and their remediation.

## 2. Results and Discussion

The grafting of mercaptosuccinic acid units onto the chitosan backbone results in the addition of a thiol functionality onto chitosan, based on a previous report [30]. The thiol-grafted chitosan scaffold was used as a matrix to bind with Ag NPs, mainly by the thiol functionality. The complementary materials characterization served to confirm the structure of the CS-SAMSA and Ag@CS-SAMSA. Since the mercaptosuccinic acid is not reactive enough to form amide bonds with the amine groups of chitosan, it was transformed to an anhydride form with excess acetyl chloride to yield an S-acetylated anhydride (SAMSA) [31]. In turn, the SAMSA was grafted onto the chitosan backbone via an amide bond and deacetylated to have free thiol groups [32]. This type of functionalized chitosan serves as a suitable matrix to coordinate Ag NPs (cf. Figure 1), whereas Appendix A outlines the synthetic route for the preparation of Ag@CS-SAMSA.

### 2.1. Characterization of the CS-SAMSA

#### 2.1.1. Viscosity Average Molecular Weight

The viscosity profile and calculation for the estimation of the viscosity average molecular weight of the prepared CS-SAMSA is shown in Appendix A. The estimated Mv value using Mark–Houwink–Sakurada (cf. Appendix A) is 112,756 g/mol.

#### 2.1.2. FT-IR Spectroscopy

The FT-IR spectra of the mercaptosuccinic acid and SAMSA are shown in Appendix A. A broad IR signature of the intermolecular hydrogen bonds of –OH functional groups at 3000–3500 cm^−1^ disappeared upon ring closure, as expected [33]. This disappearance indicates the successful formation of the anhydride structure, confirming that the –OH groups are no longer free to participate in hydrogen bonding. Meanwhile, weak IR bands near 2550 cm^−1^ related to the thiol group band were not observed [34]. This trend suggests that the thiol groups are either involved in bonding interactions or are present in concentrations that are too low to be detected. Bands near 2900 cm^−1^, which relate to C-H stretching, became more evident after the removal of a broad hydrogen bond peak due to the grafting of amine functional groups. The increased visibility of these bands further supports the successful grafting process, as it indicates the introduction of new functional groups that alter the vibrational modes of chitosan. The spectra of the SAMSA show the symmetric and asymmetric stretching of the anhydride bands at 1878 and 1680 cm^−1^, respectively [35], whereas the carboxylic group of mercaptosuccinic acid before ring closure shows a strong band at 1696 cm^−1^ [36]. The shift in these bands upon ring closure is consistent with the expected chemical transformations, highlighting the formation of the anhydride and confirming the structural modifications. The thioester functional group is noted at 1800 cm^−1^, whereas the C-O stretching of the cyclic anhydride was observable at 1357 and 1060 cm^−1^, in agreement with other reports [37,38]. These observations align with the literature values and further validate the successful synthesis and structural integrity of the SAMSA. The IR spectra of the pristine CS and its grafted form (CS-SAMSA) after deacetylation are shown in Figure 2. Both spectra show similar features, including the broad peak of the hydrogen bond at 3000–3500 cm^−1^ and bands at 2900 cm^−1^, which relates to C-H stretching; 1370 cm^−1^, which relates to –CH and –CHOH; and 1160 cm^−1^, which relates to C-O-C stretching. These consistent features indicate that the core structure of chitosan is largely preserved, even after chemical modifications. The main difference between the two bands relates to the disappearance of amide I (C=O) and amide II (N-H) at 1650 and 1590 cm^−1^, which are replaced by two new bands (1720 and 1655 cm^−1^) that relate to the free carboxylic acid and the new amide group, respectively. This shift is a critical indicator of successful grafting and deacetylation, upon the introduction of new functional groups that modify the chitosan backbone. These results provide preliminary support for the structure of the SAMSA, the grafting of the chitosan, and the deacetylation of the thioester, where the thiol functional group signature is too weak for observation in the IR spectrum. The absence of a clear thiol signature suggests that the thiol groups are either present in low quantities or are involved in bonding that alters their IR detection, reinforcing the need for complementary analytical techniques to confirm their presence.

#### 2.1.3. ^13^C CP/MAS Solid-State NMR Spectra

The structures of the CS and CS-SAMSA were evaluated using ^13^C NMR spectroscopy, and the results are illustrated in Figure 3. The spectra reveal distinct differences in various spectral signatures between the pristine and grafted chitosan. The NMR spectral features of the CS spectra consist of seven signatures that appear between 58 and 105 ppm that were attributed to the glucosamine ^13^C lines that did not show any chemical shift change before and after the grafting of the thiol groups. The chemical shift (δ) values of the ^13^C NMR signals are listed in parentheses for the various carbon lines as follows: C2–C6 (59 ppm), C3–C7 (75 ppm), C4 (82 ppm), and C1 (105 ppm). The change in the NMR line shape of the CS after grafting, as revealed by the signal broadening, was accounted for by a reduction in the crystallinity of the CS [39,40]. This broadening is indicative of successful modification, as the introduction of functional groups typically disrupts the regularity of the polymer chains, leading to decreased crystallinity. The mercaptosuccinic acid functionalization of the chitosan structure was confirmed by other complementary spectral evidence. The intensity of the ^13^C signal at 23 ppm in the CS-SAMSA spectra was enhanced compared to the pristine CS, due to the attachment of a new methylene carbon in the CS-SAMSA. In addition, a new ^13^C NMR line appeared at 38 ppm, which corresponds to a methine carbon attached to the thiol functionality of the CS-SAMSA. Moreover, an increase in the intensity of the signature at 170 ppm was attributed to the new -COOH group, which was shown in the same spectral region of the carbonyl group of the CS.

#### 2.1.4. Raman Spectroscopy

The Raman spectra of the pristine CS and CS-SAMSA are demonstrated in Appendix A. The stretching vibrations of the C-H bonds are evident at 1450–1380 cm^−1^. The band at 1260 cm^−1^ corresponds to C-N stretching, and the vibrations of C-C and C-O appear as a peak centered at 1100 cm^−1^, while the band at 950–860 cm^−1^ relates to the bending of the C-C and C-N bonds [41]. A noticeable fact before and after grafting is the increase in the intensity of the two peaks at 700 and 1650 cm^−1^, which are assigned to the deformation of carboxylic acid and the stretching of carbonyl groups, respectively, in accordance with the grafted mercaptosuccinic acid onto the CS backbone [42]. The thiol stretching signature was weak and not clearly evident in the extended Raman spectra, as revealed by a static spectrum averaged over several sample spots (cf. Appendix A). The Raman results provide evidence of the –SH group as a sharp band at 2570 cm^−1^, in accordance with the FT-IR and ^13^C-NMR spectral data [43].

#### 2.1.5. NIR Spectroscopy

Near-IR (NIR) is another spectral tool that provides further support for the presence of the thiol functionalization of the chitosan framework (cf. Appendix A). The band at 8190 cm^−1^ was attributed to the first overtone of the hydroxyl groups, while the amine and amide bands appeared at 6850 cm^−1^, where the intensity of this NIR band was sharply enhanced after grafting due to an increased number of amide functional groups [44,45]. The stretching vibrations of C-H are clearly seen at 5270 and 4060 cm^−1^. The band at 5035 cm^−1^ is a result of the first overtone band of S-H, which is seen as an intense feature in the CS-SAMSA spectrum [46]. The broad band at 4390 cm^−1^ was assigned to a combined effect of –OH stretching/bending and C-O stretching [47].

#### 2.1.6. Thermogravimetric Analysis (TGA)

The TGA profiles of the pristine and grafted chitosan are presented in Appendix A (cf. Appendix A). The thermal event near 100 °C relates to the loss of adsorbed water from the biopolymer, and the increase in area from 8.4% for the CS to 9.4% in the CS-SAMSA is accounted for by the greater abundance of hydrophilic groups in the grafted biopolymer versus the unmodified chitosan [48]. The single thermal event at 290 °C for the CS in the TGA profile was replaced by three weight loss events in the CS-SAMSA. The minor shoulder observed at 195 °C corresponds to the degradation of succinic acid, while the chitosan backbone exhibits a distinct thermal decomposition event centered around 240 °C. The lower thermal degradation temperature of the chitosan framework in the grafted biopolymer was attributed to decreased crystallinity and a reduction in hydrogen bonding upon grafting [49]. The last thermal event at 325 °C corresponds to the decomposition of the thioglycolic acid [50].

### 2.2. Characterization of the Ag@CS-SAMSA

#### 2.2.1. X-ray Photoelectron Spectroscopy (XPS)

XPS is a surface-sensitive technique that can reveal the successful deposition of Ag NPs onto CS-SAMSA, along with the oxidation state of Ag NPs. The wide and high-resolution spectra are demonstrated in Appendix A, where C*_1s_* was used as the reference. The Ag*_3d_*, C*_1s_*, N*_1s_*, O*_1s_*, and S*_2p_* binding energy peaks clearly show the presence of Ag in the CS-SAMSA matrix. Appendix A shows the high-resolution scan for Ag*_3d_*, where two major peaks at 365.5 and 371.5 eV (with a splitting of ~6 eV) relate to the binding energies of Ag*_3d5/2_* and Ag*_3d3/2_*, respectively [51]. These spectral bands were used to assess the oxidation state of the immobilized Ag NPs and to determine if any silver oxide layer formation occurred during the storage of the SERS substrate. The binding energies for Ag*_3d_* show the existence of Ag NPs in the metallic state. The deconvoluted high-resolution spectra reveal several small peaks at 367, 368, and 374 eV, which relate to the presence of minor Ag species (AgO, Ag_2_O, and Ag^0^), evident in Appendix A [52,53]. These additional peaks suggest that, while the majority of silver remains in the metallic state, a small fraction has undergone oxidation, likely due to atmospheric oxygen exposure during storage. This partial oxidation may influence the long-term stability and performance of the SERS substrate, although the dominant metallic state of the Ag NPs indicate that the overall plasmonic properties remain largely intact.

#### 2.2.2. FT-IR Spectroscopy

The characterization of Ag@CS-SAMS is well supported by the FT-IR spectral results (cf. Figure 4). The spectra of the sensor matrix before and after embedding Ag NPs were compared to study the functional groups involved in the interaction between the matrix and the Ag NPs. The intensity of the –OH groups at 3000–3500 cm^−1^ was reduced due to the role of favorable interactions between the grafted amine and carboxylic acid functional groups that coordinate with the Ag NPs, which disrupt the intermolecular hydrogen bonding of the substrate [54]. This reduction highlights the successful functionalization of the chitosan substrate, confirming the interaction between the grafted groups and the Ag NPs. The incorporation of Ag NPs was also characterized by the red shift of two strong bands at 1721 and 1664 cm^−1^, respectively [55,56]. These bands relate to the free carboxylic acid and amide functional groups that appear at 1710 and 1652 cm^−1^ after the immobilization of the Ag NPs. The red shift is indicative of the successful binding of the Ag NPs, further validating the synthetic modification (cf. Figure 1). Moreover, the characteristic Ag-S bending mode is noted at 600 cm^−1^, which supports the strong interaction of the Ag NPs with the thiol functional groups of the substrate [57]. This Ag-S bending mode confirms the presence of Ag-S bonds, which ensure the stability and effectiveness of the modified substrate in various applications.

#### 2.2.3. TEM

The morphology and size of the Ag NPs embedded in the CS-SASMA matrix were investigated by TEM and compared to the CS matrix. The TEM image in Figure 5 shows evidence of Ag NPs that are isolated and distributed homogenously in their respective matrices with variable particle size distribution. In the case of the Ag NPs in the CS matrix, the Ag particles appear more aggregated with an overall larger particle size of ca. 120 nm (cf. Figure 5a). This trend contrasts with the results for Ag@CS-SAMSA that clearly reveal the role of the added thiol and carboxylic acid functional groups in the CS-SAMSA matrix (cf. Figure 5b), where the diameter of the Ag NPs was found to be ca. 50 nm. The more homogenous distribution of the Ag NPs serves to prevent their aggregation due to the availability of S-based ligands. This well-dispersed nanoparticle distribution enhances the stability and potential effectiveness of the composite material. This difference between the pristine and modified CS is also evident during the nanoparticle synthesis, where the formation of Ag NPs in the CS dispersion yielded a brownish and cloudy solution, whereas the CS-SAMSA dispersion is pale-yellow, related to the chelation of Ag NPs with the CS-SAMSA matrix, which prevents the aggregation (cf. Appendix A).

#### 2.2.4. Raman Spectroscopy

To further evaluate the immobilization of the Ag NPs onto the CS-SAMSA, Raman spectral studies were performed in the range of 100–2000 cm^−1^, with the results being shown in Figure 6a. The signature at 225 cm^−1^ is due to bound Ag NPs with the thiol or amine functional groups, which supports that the Ag NPs are embedded in the matrix of the CS-SAMSA [58]. The remaining Raman signals of the CS-SAMSA did not change over the spectral region of 600–2000 cm^−1^ after the addition of the Ag NPs. However, there is no apparent signature for the Ag NPs due to their low content and weak Raman scattering [59]. The presence of a signature that highlights the incorporation of Ag NPs in the CS matrix reveals that chelation occurs with NPs with amine and/or hydroxyl functional groups (cf. Appendix A). This interaction is crucial for enhancing the stability and functionality of the composite material. A noteworthy observation is the quenching of the signature at 2580 cm^−1^ that arises from the thiol group after the Ag NPs’ chelation, as shown in Figure 6b, which reveals a strong interaction between -SH and the Ag NPs [60]. The same quenching process occurs for the pure CS matrix, where the amine peak at 2490 cm^−1^ disappeared after the Ag NPs’ immobilization (cf. Appendix A). The Raman spectral signature assignments in Appendix A concur with independent results from the literature.

#### 2.2.5. Two-Dimensional Raman Spectral Mapping

Since the SERS activity of the substrate mainly contributes to the local electromagnetic field of the Ag NPs, the distribution of the NPs onto the substrate was studied by 2D Raman mapping. The scanning area was 20 μm × 20 μm with an excitation wavelength at 785 nm. The Ag-S band at 225 cm^−1^ was used to perform the Raman imaging (cf. Figure 6c). The integration of the absolute area under this band was used to establish the rainbow-colored Raman image. The red areas of the image reveal regions that contain strong Ag–thiol band contributions, which also reveal that the distribution of the Ag NPs in the CS-SAMSA matrix is homogenous, in agreement with the TEM image results described above [61].

#### 2.2.6. UV-Vis Spectra of the Ag NPs on the Substrate

To maximize the SERS effect, the wavelength of excitation should match the surface plasmon resonance (SPR) frequency of the metallic NPs [62]. In Appendix A, the solid state UV-Vis spectra of Ag@CS-SAMSA are shown, which reveal that the LSPR extinction band of the Ag NPs is centered at 720 nm. This spectral position deviates significantly from the localized surface plasmon resonance (LSPR) values for the Ag NPs reported in the literature, which typically utilize a 450 nm excitation wavelength. This large difference in the LSPR band is explained by the strong coordination of Ag NPs with various thiol groups, along with the amine and hydroxyl groups of CS-SAMSA. In turn, this affects the oscillation of the free surface electrons of the metal NPs [63,64]. According to the UV-vis spectra, a 785 nm laser was used for the SERS studies.

### 2.3. SERS Performance Studies and Raman Imaging

To evaluate the SERS efficacy of the fabricated substrate, various MB dye concentrations from 1 nM to 100 μM were dropcasted onto the Ag@CS-SAMSA substrate and air-dried for 1 h. The materials were studied with a laser excitation line at 785 nm. The Raman reference spectra of the powdered MB dye were acquired prior to the SERS measurement (cf. Figure 7a). The most enhanced signal was found at 1396 cm^−1^, which can be indexed to the carbon–carbon stretching vibration of the arene rings, indicating that the MB dye was favorably adsorbed onto the substrate. This specific interaction suggests the potential for selective dye detection in mixed contaminant environments. This relates to the interaction of the MB dye cation with the Ag NPs, which contain a negative surface charge due to the reduced state of the NPs via sodium borohydride [65,66]. The characteristic SERS band is very weak for the pristine MB, whereas the spectral intensity of the MB dye onto the SERS substrate varied by increasing the dye concentration. The linear relationship observed between the dye concentration and signal intensity demonstrates the substrate’s utility for quantitative analysis. The subtracted Raman spectra are shown in Appendix A to better visualize the difference between the crystalline MB and its Raman spectra onto the SERS substrate. In addition, the same SERS study on Ag@CS for MB detection did not show signal enhancement (cf. Appendix A), whereas Ag@CS-SAMSA reveals excellent MB detection. In Appendix A, the Raman spectrum of Ag@CS-SAMSA with the deposited MB (10 μL, 10 μM) is shown under static scan conditions centered at 1396 cm^−1^.

The spatial Raman mapping of the Ag@CS-SAMSA substrate with dropcasted MB is illustrated in Figure 7b. The scanning area was 10 μm × 10 μm with an excitation at 785 nm. The highest intensity SERS signal occurred at 1396 cm^−1^ and was used to prepare the Raman image map upon integration under this spectral signature. The Raman shift at 1396 cm^−1^ was clearly observed as a red peak, which is the region with the adsorbed MB onto the Ag NPs. The corresponding Raman spectra for this red band shows a sharp feature at 1396 cm^−1^. The integrated Raman intensity is negligible for the purple area, where no associated Raman peak occurs for this region [67,68].

### 2.4. Calibration Curve and Determination of LOD and LOQ

To find the linear region between the signal strength at 1396 cm^−1^ and MB dye concentration, a calibration curve was constructed [69,70,71,72]. In Appendix A, the signal intensity showed a linear increase versus the dye concentration (on a logarithmic scale) from 1 nM to 100 μM. The slope of the calibration curve was employed to calculate the LOD (1.6 nM) and LOQ (5.6 nM), which are shown in Appendix A [73,74].

### 2.5. Dye Adsorption

The MB was adsorbed onto the surface of the Ag NPs via the N and S lone pairs via the π electrons of the electron-rich arene rings. To investigate the adsorption process and its thermodynamic aspects, common adsorption isotherms were used to analyze the SERS results [75]. Figure 8 shows the experimental SERS profile and the best fit to the Hill isotherm model, where a monolayer adsorption profile with homogeneous sites is inferred [76]. This suggests a uniform distribution of the active sites onto the Ag NPs’ surface for the MB adsorption. The best fit results of the Sips and Langmuir isotherm models (Appendix A) and the related parameters are listed in Appendix A. A comparison of these models further validates the effectiveness of the Ag NPs in achieving highly sensitive SERS-based detection.

The SERS signals reach a maxima as the MB concentration exceeds ~40 μM, which indicates the saturation of the Ag NPs. Using the Hill isotherm, the adjusted correlation coefficient (*Adj-R^2^*) is 0.98 and reveals a reliable best fit for the Hill model parameters (*K* = 19.66 and *n* = 0.35). The results show that the first layer of the adsorbed MB onto the hotspots reduce the affinity of the newly bound MB dye for the formation of the newly adsorbed layer to afford multilayer adsorption, as expected. The adsorption properties of the SERS substrate follow a monolayer profile for the MB, while there is negative cooperation between the adjacent adsorbed dye, where each spot can accommodate a single MB molecule [14].

The standard change in Gibbs energy for the adsorption (Δ*G°_ads_*) provides an indication of the affinity of the MB with the SERS substrate, which is determined by fitting Equation (4) to the SERS results to estimate *K*_eq_, before it is used to calculate Δ*G°_ads_* (−7.38 kJ/mol) by Equation (5), in agreement with another study [75]. The negative value of Δ*G°_ads_* indicates a spontaneous and favorable process for the adsorption of the MB onto the SERS substrate.

### 2.6. Enhancement Factor (EF)

To further evaluate the fabricated SERS substrate, its EF was calculated [77,78]. From a very low LOD of the SERS substrate, a high EF is expected [79]. The EF reflects the enhancement of a specific Raman signal per adsorbed molecule onto a metal NP relative to the normal Raman signal per molecule in the absence of NPs [80,81]. The average EF is obtained by Equation (1).
(1)$$EF=\frac{I_{SERS}}{I_{bulk}}\;\times \;\frac{N_{bulk}}{N_{SERS}}$$

*I_SERS_* and *I_bulk_* are the intensity of SERS and the Raman vibrational band at 1396 cm^−1^, respectively. *N_SERS_* is the number of MB dye molecules in the zone of the laser (785 nm) that yield the signal in the focal volume of the beam on the SERS substrate and *N_bulk_* is the same parameter in a single crystal of the target molecule. *N_SERS_* includes all the MB molecules in the laser spot, not the adsorbed species onto the Ag NPs, where the real value of the EF is lower than the reported value. The ratio of analyte on the SERS substrate and in the bulk is obtained by Equation (2):(2)$$\frac{N_{bulk}}{N_{SERS}}=\frac{2344\lambda }{{NA}^{2}}\times A\times \frac{\rho }{w}$$
where *λ* is the wavelength of the applied laser light (nm), *NA* is the numerical aperture of the lens employed, *A* is the substrate area that was covered by the MB solution (cm^2^), *ρ* is the density of MB crystal (g/cm^3^), and *w* is the mass of MB solution distributed onto the SERS substrate (ng) [82]. Employing Equation (1) above, the intensity of the Raman band at 1396 cm^−1^ was enhanced by a factor of ca. 1.3 × 10^8^ (cf. the sample EF calculation in Appendix A), which indicates a superior performance of the fabricated substrate [83,84]. By comparison, the enhancement factor (EF) is reported for several cationic dyes (MB, rhodamine 6G, and rhodamine B) and an acidic dye (rose Bengal) in Appendix A. The EF values for the various dye systems vary from 5 × 10^4^ to 2.6 × 10^8^, where the value obtained for the Ag@CS-SAMSA sensor materials with MB is the highest. The limit of detection (LOD) varies from 10^0^ to 10^4^ nM for the various dyes, whereas the LOD for the MB (1.6 nM) compares favorably with the best substrates listed in Appendix A, such as graphene or chitosan foams that contain Ag NPs. Based on the sustainability of chitosan and its synthetic versatility, the EF and LOD reported for the Ag@CS-SAMSA sensor materials for the detection of MB highlights the outstanding results obtained herein. This is further supported by the facile substrate synthesis and its favorable dye detection of MB, as compared with the other SERS substrate dye systems listed in Appendix A.

### 2.7. Orientation of the Probe Molecules

The functional groups of chromophores define their relative affinity to the NPs, which affect the intensity of the SERS signal [85]. Lone pairs and electron-rich aromatic rings will bond to metallic NPs through London dispersion forces, where such weak intermolecular interactions (ca. 2–6 kcal/mol) can yield a SERS effect [86]. Parallel adsorption geometry via π electrons promote the red shift of Raman signatures, as reported elsewhere [87]. The Raman band centered at 1396 cm^−1^ is attributed to ring stretching and red shifts after the π interaction with the Ag NPs. Herein, the parallel adsorption geometry was adopted, as illustrated in Appendix A.

The Stokes Raman shift of the ring bands of the planar adsorbed arene-based systems relative to the bulk spectra can be accounted for by the back donation of the electron density from the π* antibonding orbitals [88]. This shift is illustrated in Appendix A.

### 2.8. Distance Dependence of SERS Signals

It is worth noting that the SERS intensity is strongly dependent on the distance between the dye molecule and the metal NPs [89]. In other words, the SERS effect is observable when the target analyte resides in the plasmon resonance field region created by the metal NPs [90,91]. The SERS intensity (*I_SERS_*) is inversely proportional to the distance between the target molecule and the NPs, which decays rapidly with greater distance from the surface of the NPs, along with the radius of curvature for the NPs [92,93]. The functional dependence of the spectral intensity is described by Equation (3).
(3)$$I={\left(1+\frac{r}{a}\right)}^{-10}$$

Herein, the value of *I* is the relative SERS signal at 1396 cm^−1^ for MB (10 μM) at the highest and lowest intensity, *a* is the radius of curvature of the Ag NPs, and *r* is the distance between the target molecule and the metal NP surface. The radius of curvature for the Ag NPs was estimated from TEM (25 nm). The fabricated SERS substrate contains Ag NPs on the surface or embedded within the matrix of the modified biopolymer. The calculation shows the distance of the MB dye for a deeply embedded Ag NP is less accessible versus a metal NP at the substrate surface that can bind with the MB dye. The distance of the adsorbate for a particle positioned on the surface is estimated to be a maximum distance of 1.3 nm (cf. Appendix A).

### 2.9. Effect of Ag NP Concentration

Generally, the SERS performance relates to the concentration of Ag NPs in the SERS substrate [94]. To establish the optimized concentration of Ag NPs in the SERS substrate, three substrates with variable loadings of Ag NPs were prepared and used to detect the MB dye. In Appendix A, the Raman data for 50, 150, and 1500 mg of Ag NPs per gram of substrate are displayed. Notably, the intensity of the band at 1396 cm^−1^ was varied from (3–5) × 10^3^ when the concentration of Ag NPs increased from 50 to 150 mg/g, whereas a ten-fold increase in concentration (ca. 1500 mg/g) does not affect the 1396 cm^−1^ band considerably. According to these spectra, a concentration of 150 mg/g for the Ag NPs was selected as the optimized level of the metallic NPs for the analysis of the SERS substrate.

### 2.10. Regeneration, Reproducibility, and Storage Stability of the Substrate

The regeneration and reuse of the SERS substrate is essential from both an economic and environmental viewpoint [95,96]. To evaluate the regeneration of Ag@CS-SAMSA loaded with MB as the spent substrate, the material was washed with methanol and deionized water to desorb the MB from the substrate. After each washing step, the removal of the MB was monitored via dye decolorization from the substrate, in comparison to the dark grey appearance before the MB addition (cf. Appendix A). The reuse of the washed substrate was repeated for three cycles. The SERS intensity (*I_SERS_*) decreased by 3.1% in the second cycle and 6.3% in the third cycle. Although the SERS intensity was slightly reduced over three regeneration cycles, the favorable SERS activity that is retained for Ag@CS-SAMSA highlights its promising utility for the cyclic detection of MB even at a low concentration.

Reproducibility is an important parameter that requires consideration for the fabrication of a reliable SERS substrate [79]. The results of the SERS detection of MB at two selected sites from five independent batches of the substrate were recorded, as illustrated in Appendix A. The relative standard deviation (RSD) of the SERS signal intensities show a superior consistency (RSD = 5.2%) at 1396 cm^−1^, revealing the uniformity of the substrate. Previous work indicates that RSD values below 15% are acceptable for different SERS substrates [97].

One drawback of SERS substrates that contain Ag NPs is their relative instability against oxidation. The silver oxide layer on the surface of the NPs reduces their SERS activity that affects the substrate performance. Thus, the long-term stability of the SERS substrate is an important parameter that was investigated herein over one month by measuring the SERS spectra of MB at 10 μM. Appendix A reveals the intensity of the SERS peak at 1396 cm^−1^ versus the storage time, where the SERS intensity shows reliability over ten days, whereas the SERS intensity dropped to 5% of its initial value after 30 days. The results strongly support that the Ag NPs are stabilized by the CS-SAMSA matrix from atmospheric oxygen at ambient conditions, along with the prevention of the aggregation of NPs, which highlight the favorable stability of the biopolymer substrate.

## 3. Materials and Methods

### 3.1. Materials

Low molecular weight chitosan (~75–85% deacetylation), mercaptosuccinic acid, acetyl chloride, pyridine, hydroxyl amine, silver nitrate, sodium borohydride, dipotassium hydrogen phosphate, potassium dihydrogen phosphate, diethyl ether, and acetone were purchased from Sigma-Aldrich and were used without further purification. Millipore water was used for the preparation of the aqueous samples.

### 3.2. Methods

#### 3.2.1. Synthetic Methods

The synthesis of the biocomposite that contains CS-SAMSA and Ag nanoparticles (NPs), denoted as Ag@CS-SAMSA, including the deposition of the SERS substrate by dropcasting onto a glass microscope slide, is detailed in the Methods section of the Appendix A.

#### 3.2.2. Characterization Methods

The viscosity average molecular weight (M_v_) of the CS-SAMSA was estimated according to a method reported by Xue and Wilson [98] by estimating the intrinsic viscosity (η_i_) of each sample by Equation (S1) [99,100] in the Appendix A. The viscosity average molecular weight is calculated by using the Mark–Houwink–Sakurada equation, Appendix A, where α and K are constants (K = 1.81 × 10^−6^ L/g and α = 0.93) for the solvent system employed [101]. The degree of acetylation of the chitosan was determined using ^1^H NMR spectroscopy, as described in the literature [102].

The spectral (IR, solid-state ^13^C NMR, Raman, X-ray photoelectron, and near-IR), thermogravimetry (TG), microscopy (TEM), and surface-enhanced Raman spectroscopy (SERS) characterizations were reported previously, [103] and the specific details are outlined in Appendix A. The Raman excitation wavelength employed was 785 nm.

#### 3.2.3. Adsorption Studies, Isotherm, and Modelling

The intensity of the Raman band at 1396 cm^−1^ was recorded at various MB dye concentrations onto the SERS substrate by using the data acquired from the SERS measurements. The equilibrium isotherm was plotted by the SERS intensity (*I_SERS_*) vs. *C* (MB dye concentration) and then analyzed using the Hill model (Equation (4)) [12,104,105,106].
(4)$$I_{SERS\;}=I_{SERS}^{max\;}\;\frac{C^{n}}{K_{eq}+C^{n}}$$

Here, *I_SERS_* is the Raman intensity representing an equilibrium condition, *C* is the concentration of MB (nM), *I_SERS_^max^* is the maximum Raman intensity, *K_eq_* is the equilibrium (Hill isotherm) constant (L/nM) for the adsorption process of MB onto the solid SERS substrate, and *n* is the cooperative binding constant. For cases where *n* > 1 or *n* < 1, attraction or repulsion can occur between the adsorbates. When *n* = 1, the Hill isotherm describes the adsorption profile that conforms to the Langmuir model.

The standard change in the Gibbs energy of adsorption (Δ*G°_ads_*) was quantified by an equilibrium adsorption constant (*K_eq_*), according to Equation (5):(5)$${G{}^{\circ}}_{ads}=-RTlnK_{eq}$$
where *R* is the ideal gas constant (8.314 J/mol.K) and *T* is the temperature in K [32].

## 4. Conclusions

In this study, a novel chitosan-based SERS substrate impregnated with Ag NPs was prepared and employed as an effective and cost-efficient sensor for the detection of MB (methylene blue) over a wide dye concentration range, marking the first use of S-acetyl mercaptosuccinic acid (SAMSA) for the functionalization of chitosan to achieve a significantly higher sensitivity compared to existing methods and SERS sensor materials (cf. Appendix A). The study of the structure–function properties of the biocomposite via Raman scattering experiments with MB as the dye probe highlight an EF~10^8^ with an LOD of 1.6 nM. The substrate demonstrates favorable long-term storage stability and consistent spectral reproducibility for quantitative SERS applications. Two-dimensional Raman imaging results reveal a uniform distribution of Ag NPs on the substrate, where the MB reveals extensive enhancement near the surface of the Ag NPs for interaction with these domains. The MB adsorption onto the substrate follows the Hill isotherm model, where the monolayer and non-cooperative adsorption occur between the Ag NPs and MB, favored by the cation arene dye structure. The level of Ag NPs in the substrate was optimized at 150 mg/g by studying the SERS response. We demonstrate the innovative design of a unique SERS substrate with promising features for the analytical detection of MB in aqueous media at remarkably low (nM) levels. The biocomposite described here displays distinctive dual-function capabilities (adsorption and sensing), positioning it as a transformative material in the development of sustainable SERS systems. Future work will focus on several key areas that build upon the promising results of the Ag@CS-SAMSA biocomposite sensor from this study, optimizing the SERS substrate synthesis by the variable grafting of chitosan and the immobilization of Ag NPs that can enhance the sensor’s stability and sensitivity. The enhancement of the substrate performance should be complemented by studies across a wider array of detectable contaminants, which is crucial for improving the sensor’s versatility for practical applications, especially for environmental water samples. In turn, such hybrid sensor materials hold promising potential for advanced applications, such as biomedical devices, environmental monitoring, and water treatment technologies.

## Figures and Tables

**Figure 1 ijms-25-09327-f001:**
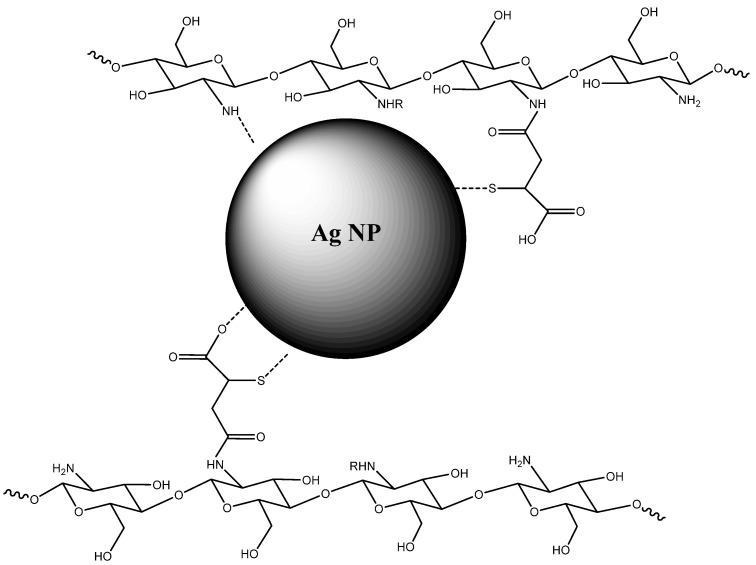
An illustrative view of a fragment of the chemical structure of Ag@CS-SAMSA, which shows the coordination of Ag NPs by thiol-based groups of the grafted chitosan.

**Figure 2 ijms-25-09327-f002:**
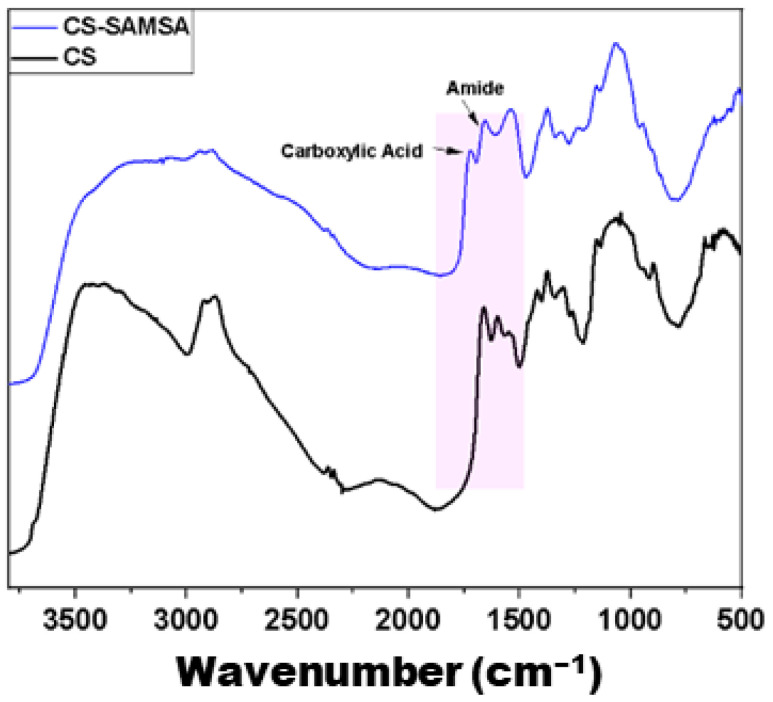
Chemical structure of Ag@CS-SAMSA, which shows the coordination of Ag NPs by functional groups of the grafted chitosan.

**Figure 3 ijms-25-09327-f003:**
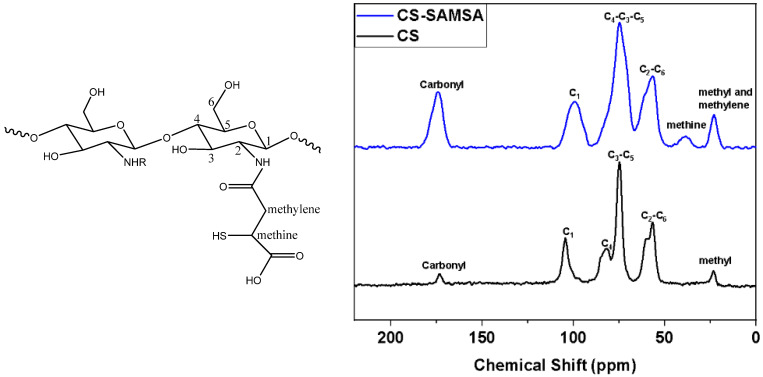
^13^C CP/MAS NMR spectra of CS (black) and CS-SAMSA (blue). For the numbering scheme, refer to the assigned chemical structure shown in the inset above.

**Figure 4 ijms-25-09327-f004:**
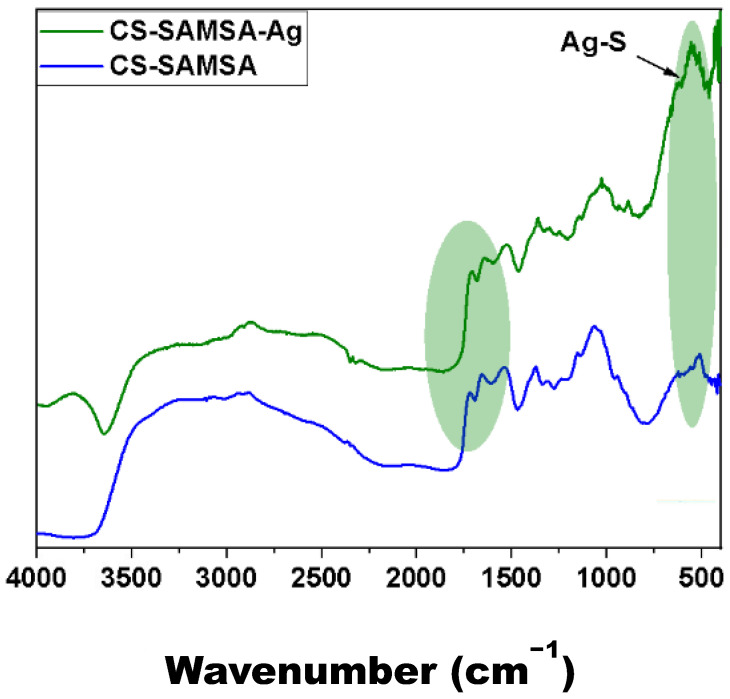
FT-IR spectra of CS-SAMSA (blue) and Ag@CS-SAMSA (green). The highlighted region shows the main spectral changes after the incorporation of Ag NPs onto CS-SAMSA.

**Figure 5 ijms-25-09327-f005:**
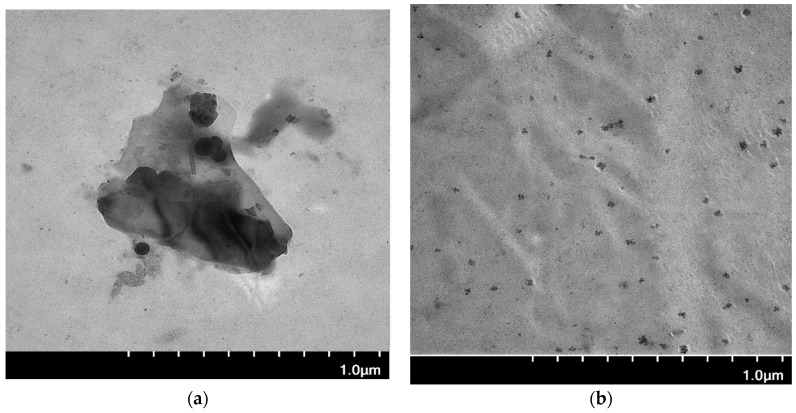
TEM images of composites that contain Ag NPs with CS or CS-SAMSA: (**a**) Ag@CS, and (**b**) Ag@CS-SAMSA.

**Figure 6 ijms-25-09327-f006:**
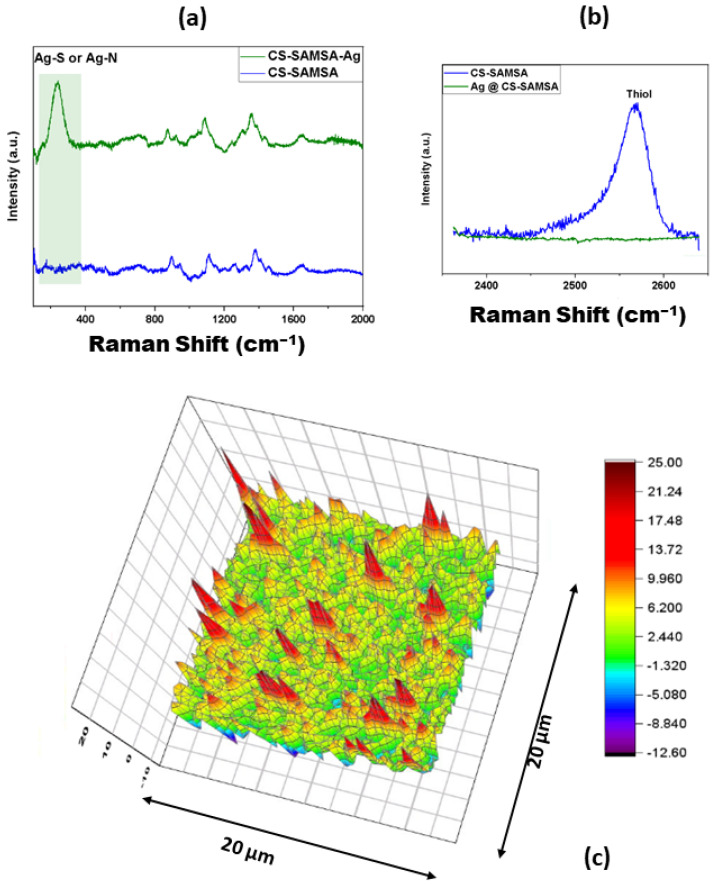
(**a**) FT-Raman spectra of CS-SAMSA (blue) and Ag@CS-SAMSA (green) with the highlighted region showing the main difference after the addition of Ag NPs; (**b**) FT-Raman spectra of CS-SAMSA (blue) and Ag@CS-SAMSA (green), with a static scan centered at 2550 cm^−1^; and (**c**) Raman imaging result of Ag@CS-SAMSA, where the Raman spectra were obtained under static scan conditions centered at 225 cm^−1^ with a laser excitation wavelength of 785 nm.

**Figure 7 ijms-25-09327-f007:**
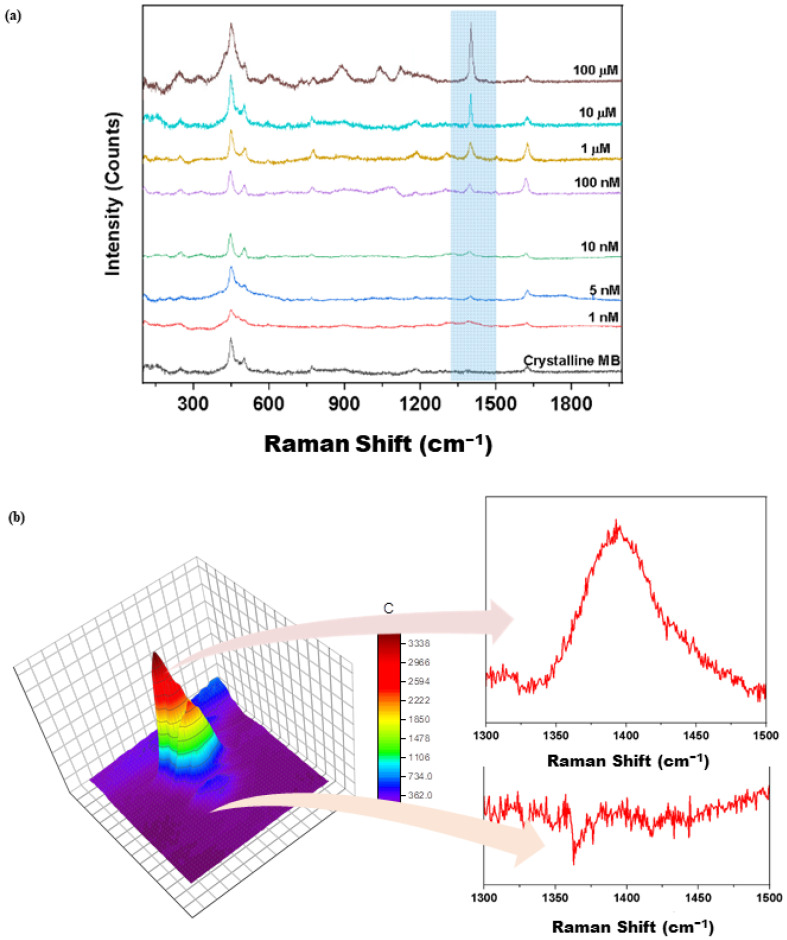
(**a**) SERS spectra of different concentrations of the MB on Ag@CS-SAMSA. The lowest spectrum is crystalline MB powder that was compared to reveal the effect of SERS substrate. (**b**) Raman imaging result of Ag@CS-SAMSA, where the spectrum was obtained under static scanning at 1396 cm^−1^ with a fixed laser excitation (λ_ex_ = 785 nm). The left Raman spectrum indicates the red region is a domain rich in MB, while the purple region does not include MB.

**Figure 8 ijms-25-09327-f008:**
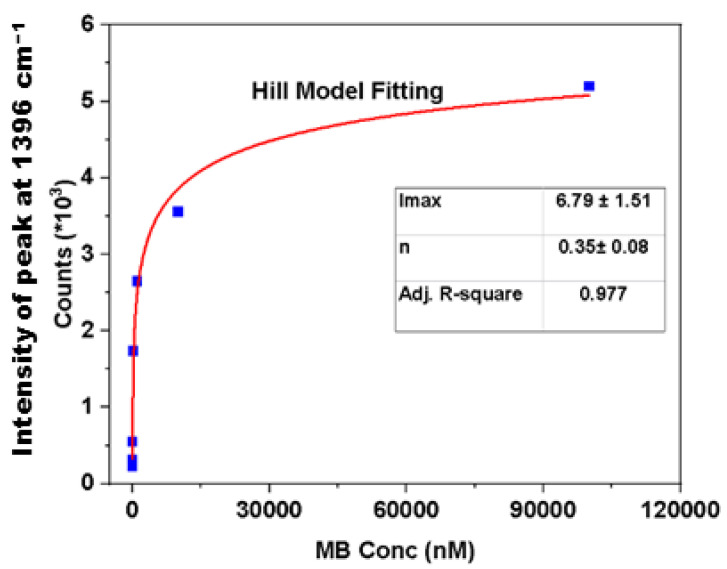
Adsorption isotherm of MB on SERS substrate fitted by the Hill model, where the MB concentration varies from 1 nM to 100 μM for a 1 h contact time at 295 K.

## Data Availability

The raw data supporting the conclusions of this article will be made available by the authors on request.

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
