# Peer review of "A Highly Sensitive Chitosan-Based SERS Sensor for the Trace Detection of a Model Cationic Dye"

_ijms, 2024, doi:10.3390/ijms25179327_

Round 1

Reviewer 1 Report

Comments and Suggestions for Authors This manuscript must be significantly improved based on the following issues:   1.- The authors must improve the English grammar and style of all the sections of the manuscript. 2.- The abstract section is not clear. This section should include introduction, materials and methods, main results, and conclusion. 3.- The introduction section must incorporate the research problem to resolve. In addition, this section must add the advantages and limitations of the works proposed in the literature to resolve the research problem.  4.- The introduction section must consider the innovation or scientific contribution and advantages of the proposed sensor in comparison with others reported in the literature. 5.-The second section must be materials and methods. The third section must be results and discussion. 6.- The authors must include more detailed description of the materials and methods. Furthermore, all the parameters used in the equations must be described. 7.- In the section of materials and methods, the authors should include figures or schematically views to improve the description of materials and methods. 8.- The third section must be results and discussion. This section must be significantly improved. The authors must include more critical discussions of the behavior of the results reported in the Figures 2-8. 9.- The authors must improve the resolution of the Figures 2, 3, 4, and 8. The discussion of the behavior of the results of Figures 2, 3, and 4 must be improved. 10.- The authors must add more discussions of the behavior of the results of Figures 6, 7, and 8. 11.- The authors should include a table with the main parameters, advantages, and drawbacks of the proposed sensor in comparison with others reported in the literature. 12.-The authors must incorporate the challenges of the proposed sensor. 13.-What are the future research works? 14.-The conclusion must be significantly improved based on the above comments. Comments on the Quality of English Language

The authors must improve the English grammar and style of all the sections of the manuscript.

Author Response

Response to Reviewer Comments on MS ID:  ijms-3163738 for manuscript entitled “A highly sensitive chitosan-based SERS sensor for trace detection of a model cationic dye”

The authors appreciate the valuable comments provided by the reviewers. Outlined below are the responses to the reviewer comments in red font.

Reviewer #1

This manuscript must be significantly improved based on the following issues:  

 1.- The authors must improve the English grammar and style of all the sections of the manuscript.

Response: The revised manuscript was revised to address the English grammar and style.

2.- The abstract section is not clear. This section should include introduction, materials and methods, main results, and conclusion.

Response: The abstract was revised to address the reviewer comment.

3.- The introduction section must incorporate the research problem to resolve. In addition, this section must add the advantages and limitations of the works proposed in the literature to resolve the research problem. 

Response: The recommended comment related to the research problem was added. The last paragraph of the Introduction was revised to address the reviewer comments.

4.- The introduction section must consider the innovation or scientific contribution and advantages of the proposed sensor in comparison with others reported in the literature.

Response: The features of the proposed sensor is mentioned in the introduction, along with a literature comparison, as suggested by the reviewer.

5.-The second section must be materials and methods. The third section must be results and discussion.

Response: We have been instructed by the Editor that the current format is in accordance with IJMS template and the author guidelines so the existing format will be retained.

6.- The authors must include more detailed description of the materials and methods. Furthermore, all the parameters used in the equations must be described.

Response: The materials & methods are described in detail with appropriate citations, as needed.

7.- In the section of materials and methods, the authors should include figures or schematically views to improve the description of materials and methods.

Response: The requested figures were added to Supporting Material.

8.- The third section must be results and discussion. This section must be significantly improved. The authors must include more critical discussions of the behavior of the results reported in the Figures 2-8.

Response: The updated discussion for Figures 2 to 8 are presented in the revised manuscript.

9.-The authors must improve the resolution of the Figures 2, 3, 4, and 8. The discussion of the behavior of the results of Figures 2, 3, and 4 must be improved.

Response: Figures 2-4 have been improved as recommended.

10.- The authors must add more discussions of the behavior of the results of Figures 6, 7, and 8.

Response: Additional discussion of the results for Fig. 6-8 have been revised, as recommended.

 11.- The authors should include a table with the main parameters, advantages, and drawbacks of the proposed sensor in comparison with others reported in the literature.

Response: The response to this query is addressed in Table S3.

12.-The authors must incorporate the challenges of the proposed sensor.

Response: The challenges of the proposed sensor was described in Section 2.1.

13.-What are the future research works?

Response: The reviewer query was addressed in the final section of the Results and Discussion section.   

14.-The conclusion must be significantly improved based on the above comments.

Response: The conclusion was edited to address the reviewer comment.

In summary, we wish to acknowledge the insightful and constructive comments provided by Reviewer #1, along with the opportunity to improve the quality of the manuscript submission. We have also carried out language editing to address the clarity and syntax of the revised manuscript to meet the high standards of this journal.

Reviewer 2 Report

Comments and Suggestions for Authors

Bahareh et al. presented a well-written study on using an Ag@CS-SAMSA matrix for SERS detection of methylene blue (MB) at very low concentrations down to 1.6 nM. The authors also addressed the reproducibility and stability of the SERS substrate. However, there are a few recommendations the authors might consider before publication.

#To enhance the manuscript, it would be beneficial to emphasize the unique advantages of using SAMSA for chitosan grafting in SERS applications. This will help to distinguish your study from other similar works, particularly those employing common grafting agents like cinnamaldehyde and others. By highlighting the novelty of your approach, you can better demonstrate the originality and significance of your research.

#Kindly specify the Raman excitation wavelength at the beginning of the Raman discussion in the experimental section to provide clarity. 

#Few results, such as the XPS and UV-Vis data, would benefit from further explanation. Explaining these findings could offer deeper insights into the study’s outcomes.

Author Response

Response to Reviewer Comments on MS ID:  ijms-3163738 for manuscript entitled “A highly sensitive chitosan-based SERS sensor for trace detection of a model cationic dye”

The authors appreciate the valuable comments provided by the reviewers. Outlined below are the responses to the reviewer comments in red font.

Reviewer #2

Bahareh et al. presented a well-written study on using an Ag@CS-SAMSA matrix for SERS detection of methylene blue (MB) at very low concentrations down to 1.6 nM. The authors also addressed the reproducibility and stability of the SERS substrate. However, there are a few recommendations the authors might consider before publication.

#To enhance the manuscript, it would be beneficial to emphasize the unique advantages of using SAMSA for chitosan grafting in SERS applications. This will help to distinguish your study from other similar works, particularly those employing common grafting agents like cinnamaldehyde and others. By highlighting the novelty of your approach, you can better demonstrate the originality and significance of your research.

Response: We have addressed the novelty of the work as outlined in the revised manuscript.

#Kindly specify the Raman excitation wavelength at the beginning of the Raman discussion in the experimental section to provide clarity. 

Response: Added

#Few results, such as the XPS and UV-Vis data, would benefit from further explanation. Explaining these findings could offer deeper insights into the study’s outcomes.

Response: The XPS & UV-vis results have been further explained to address the reviewer query.

In summary, we wish to acknowledge the insightful and constructive comments provided by Reviewer #2, along with the opportunity to improve the quality of the manuscript submission. We have also carried out language editing to address the clarity and syntax of the revised manuscript to meet the high standards of this journal.

Round 2

Reviewer 1 Report

Comments and Suggestions for Authors

The manuscript has been significantly improved and presents a promising research proposal that could potentially make a significant impact in the field. 

However, the authors should they must make minor changues and improve their presentation of the manuscript in order to be accepted 

This manuscript must be improved based on the following issues:

In general, the paper is poorly organized. It should be organized correctly, the images corrected, and the results discussed. The authors should also compare their results obtained with respect to related research. 

It is strongly recommended that authors adhere to the IMR&D methodology, as it will provide a solid framework for their research. 

(Introduction, Methodology, Results, and Discussion).  

Conclusions 

The images should be of better quality and should not be overlapping.

The authors' FTIR images could be better edited (apparently, there are two images), and they should explain why they used the absorbance mode. 

The authors' SEM micrographs need to be completed and presented better. 

the authors should compare their results with related research and discuss and conclude their results. 

The section on future work should be integrated into the conclusion, ensuring that the conclusion fully appreciates and understands the authors' work. 

Comments on the Quality of English Language

its ok 

Author Response

Response to Reviewer Comments on MS ID:  ijms-3163738 for manuscript entitled “A highly sensitive chitosan-based SERS sensor for trace detection of a model cationic dye”

The authors appreciate the valuable comments provided by the reviewers. Outlined below are the point-by-point responses to the reviewer comments in red font.

Reviewer #1

This manuscript must be significantly improved based on the following issues:  

The manuscript has been significantly improved and presents a promising research proposal that could potentially make a significant impact in the field. 

 However, the authors should they must make minor changues and improve their presentation of the manuscript in order to be accepted.

This manuscript must be improved based on the following issues:

In general, the paper is poorly organized. It should be organized correctly, the images corrected, and the results discussed. The authors should also compare their results obtained with respect to related research. 

Response: The order of sections is in accordance with IJMS template. This organization is confined by the nature of the template for authors.

Table S3 compares the results of this research with other similar reported sensors in the literature for several cationic dyes (methylene blue, Rhodamine 6G, Rhodamine B) and an acidic dye (rose Bengal). A comparison of the EF and LOD is presented in the final paragraph of introduction and Section 2.6

It is strongly recommended that authors adhere to the IMR&D methodology, as it will provide a solid framework for their research. 

(Introduction, Methodology, Results, and Discussion).  

Conclusions 

Response: The order of sections is in accordance with IJMS template, where the methodology precedes the conclusion. Refer to the following URL:  IJMS | Instructions for Authors (mdpi.com)

The images should be of better quality and should not be overlapping.

Response: Corrections were applied to address the reviewer comment.

The authors' FTIR images could be better edited (apparently, there are two images), and they should explain why they used the absorbance mode.

Response: The spectra of the sensor matrix before and after embedding Ag NPs were compared to study the functional groups involved in the interaction between the matrix and Ag NPs. FTIR was primarily used here to analyze these functional groups, making a more reliable quantitative comparison of the spectral results.

The rationale for the use of absorbance mode was added to the Section S3 of the Supporting Materials, where all the instrumentation were explained in details. 

The authors' SEM micrographs need to be completed and presented better.

Response: We have not use SEM in this manuscript. We contend that the 2D Raman imaging results provide similar information (cf. Figure 6 & Fig. S18). As well, 2D imaging provided chemical information according to the spectral signatures employed, as described in another study (doi:10.1016/j.carbpol.2018.10.111).

The authors should compare their results with related research and discuss and conclude their results. 

Response: Table S3 in the Supporting Material (SM) provides a comparison of the results of this research with other similar reported SERS-based sensors in the literature. A comparison also added to the last paragraph of introduction and Section 2.6.

The section on future work should be integrated into the conclusion, ensuring that the conclusion fully appreciates and understands the authors' work. 

Response: The future work was integrated into the latter part of the conclusion section, as recommended by the reviewer, and updated accordingly.

In summary, we wish to acknowledge the insightful and constructive comments provided by Reviewer #1, along with the opportunity to improve the quality of the manuscript submission. We have also carried out language editing to address the clarity and syntax of the revised manuscript to meet the high standards of this journal.
